# Mechanisms Underlying the Suppression of Chromosome Rearrangements by Ataxia-Telangiectasia Mutated

**DOI:** 10.3390/genes12081232

**Published:** 2021-08-10

**Authors:** Motohiro Yamauchi

**Affiliations:** Hospital Campus Laboratory, Radioisotope Center, Central Institute of Radioisotope Science and Safety, Kyushu University, Fukuoka 812-8582, Japan; yamauchi.motohiro.619@m.kyushu-u.ac.jp; Tel.: +81-92-642-6194

**Keywords:** chromosome rearrangement, ataxia-telangiectasia mutated, DNA double-strand breaks

## Abstract

Chromosome rearrangements are structural variations in chromosomes, such as inversions and translocations. Chromosome rearrangements have been implicated in a variety of human diseases. Ataxia-telangiectasia (A-T) is an autosomal recessive disorder characterized by a broad range of clinical and cellular phenotypes. At the cellular level, one of the most prominent features of A-T cells is chromosome rearrangement, especially that in T lymphocytes. The gene that is defective in A-T is ataxia-telangiectasia mutated (*ATM*). The ATM protein is a serine/threonine kinase and plays a central role in the cellular response to DNA damage, particularly DNA double-strand breaks. In this review, the mechanisms by which ATM suppresses chromosome rearrangements are discussed.

## 1. Introduction

Ataxia-telangiectasia mutated (ATM) plays a central role in the cellular responses to DNA damage, particularly DNA double-strand breaks (DSBs) [1]. ATM is a serine/threonine kinase belonging to a family of phosphoinositide 3-kinase-related kinases, and ataxia-telangiectasia mutated and rad3-related (ATR) and catalytic subunit of DNA-dependent protein kinase (DNA-PKcs) also belong to the same protein family [2]. In response to DSBs, ATM kinase is activated and phosphorylates proteins involved in cell cycle checkpoint regulation and DNA repair, thereby initiating the response to DNA damage [2]. Consistent with its key role in maintaining genome integrity, germline mutations in *ATM* cause the congenital disease, ataxia-telangiectasia (A-T) [1]. A-T patients exhibit a broad range of clinical characteristics, including cerebellar ataxia, oculocutaneous telangiectasia, immunodeficiency, neurological degeneration, premature aging, radiation hypersensitivity, and predisposition to hematopoietic malignancy and cancer [3]. At the cellular level, chromosome instability is one of the most common features of A-T [4,5]. 

Chromosome rearrangement (CR) refers to structural variations (SVs) in chromosomes, such as inversions and translocations [6]. CRs have been implicated in a variety of human diseases and conditions, including intellectual disability, autism, and cancer [6]. It is well known that cells obtained from A-T patients, particularly lymphocytes, show the presence of damaged chromosomes and CRs at a higher frequency than those from healthy individuals [4,5].

Here, findings from different studies on CR and ATM are reviewed, and the mechanisms by which ATM suppresses CR are discussed.

## 2. Chromosome Rearrangement (CR)

### 2.1. Types of CR

CR refers to SVs in chromosomes [6]. CR includes intrachromosomal SVs, such as inversions, duplications, and deletions, or interchromosomal SVs, such as translocations and dicentrics (Figure 1) [6]. Duplications and deletions lead to copy-number variations (CNVs) in genes, whereas inversions and (balanced) translocations are copy-neutral. Such CRs can be detected using chromosome banding, fluorescence in situ hybridization (FISH), and array comparative genome hybridization (CGH). Array CGH can be used to detect CNVs formed as a result of deletions and duplications. Chromosome banding and FISH can be used to detect copy-neutral SVs, such as inversions and translocations. Recent advances in sequencing techniques have revealed more complex types of CRs, e.g., chromothripsis and chromoanasynthesis (Figure 2) [7,8,9]. Chromothripsis is defined as a complex rearrangement resulting from a catastrophic event in which one or a few chromosomes are shattered and rejoined randomly (Figure 2a) [7,8,9]. Chromoanasynthesis is a type of complex rearrangement that occurs by errors during replication [7,8,9]. Both chromothripsis and chromoanasynthesis are believed to occur during a single cell cycle [9]. The two types of highly-localized complex rearrangements are collectively termed as “chromoanagenesis” [7,8].

### 2.2. Mechanisms of CR

#### 2.2.1. DNA Damage Leading to CR

CR requires the concomitant occurrence of two or more DSBs at different genomic locations [10]. Therefore, DSB is the major type of DNA lesion that causes CR.

#### 2.2.2. Sources of DSBs

##### Physiological Sources of DSBs

V(D)J Recombination and Class Switch Recombination (CSR).

Immune cells, such as B and T cells, have two mechanisms for the introduction of DSBs into antibody genes: V(D)J recombination and Ig CSR (Figure 3a,b). Genes encoding Igs and T-cell receptors contain exons for antigen-binding domains. The exons encoding the antigen-binding domains are assembled from variable (V), diversity (D), and joining (J) gene segments (Figure 3a) [11]. The V, D, and J segments contain multiple exons. V(D)J recombination is initiated by the two enzymes RAG1 and RAG2 [11]. These enzymes introduce DSBs at conserved DNA sequence elements, termed recombination signal sequences, which are adjacent to each V, D, and J coding segment. Following this, each exon of the V, D, and J segments is rejoined (Figure 3a) [11]. Numerous combinations of the rejoining among each exon of the V, D, and J segments increase the diversity of antigen receptors. CSR is a mechanism by which B cells switch classes from IgM to IgG, IgA, or IgE (Figure 3b) [12]. In CSR, activation-induced cytidine deaminase (AID) catalyzes cytosine deamination in the S region of the constant regions in Ig heavy chain genes [12]. Cytosine deamination induces the conversion of cytosine to uracil [12]. AID-generated uracil is removed by base excision repair (BER) or mismatch repair (MMR) [12]. Single-strand breaks (SSBs) arise during BER and MMR, which can spontaneously form DSBs if they are situated close to each other on opposite DNA strands [12].

###### Transcription

Several studies have reported that CRs occur preferentially in transcribed regions in both human cancer and mouse models [13,14,15]. Accumulating evidence shows that transcription is a natural source of mutations and rearrangements from bacteria to humans [16]. Non-B DNA structures formed transiently during transcription, such as RNA-DNA hybrids, G-quadraplexes, hairpins, or cruciform DNA, are more susceptible to nuclease attack, which can lead to the introduction of transcription-associated DSBs [16]. During transcription, torsional stresses are exerted on DNA, owing to the action of RNA polymerase [16,17]. Positive or negative supercoiling arises ahead of or behind RNA polymerase, respectively [16,17]. Torsional stress is relieved by topoisomerases 1 and 2 (Top1 and Top2) [16,17,18]. Top1 introduces SSBs in DNA and relieves negative supercoiling behind RNA polymerase [16]. In contrast, Top2 introduces DSBs and relaxes positive supercoils generated ahead of RNA polymerase [16]. The action of Top1 or Top2 involves the introduction of SSBs or DSBs, covalent binding of the enzyme to the broken DNA ends, resealing of the breaks, and dissociation of the enzyme from DNA [16,17,18]. If the enzymes fail to complete these reactions, the topoisomerase cleavage complex (Top1cc or Top2cc), a structure in which DNA breaks are covalently bound by Top1 or Top2, is formed [16,17,18]. There is ample evidence that certain hormones cause DSB formation in a manner dependent on transcription and topoisomerase 2β (Top2β) expression [19,20,21,22]. Several groups have reported that estrogen, a sex hormone predominantly functional in females, introduces DSBs in the promoter regions of estrogen-responsive genes [19,20,21]. DSB induction was found to depend on the expression of estrogen receptor α (ERα) and Top2β [19,20,21]. Stork et al., demonstrated that CRs are enriched in estrogen-responsive genes [21]. The Top2β-dependent DSB induction observed in ERα-dependent promoters was also observed in the promoters depending on androgen receptor, retinoic acid receptor, and thyroid hormone receptor, indicating that Top2β-dependent DSB induction represents a common strategy for regulated gene transcription [17,19,22]. 

###### Replication

DNA lesions on the template strand are known to introduce DSBs during replication [23,24,25]. The DNA lesions include SSBs and oxidative base damage, such as 8-oxoguanine and thymine glycol [23,24]. These single-strand lesions occur spontaneously [26]. Single-strand lesions are estimated to be introduced at an approximate rate of 1000 per cell per hour [24]. Although the vast majority of these lesions are rapidly repaired via dedicated repair pathways, such as SSB repair or BER, some of the lesions remain unrepaired and cause replication fork stalling, and in more serious cases, fork collapse, which leads to the generation of DSBs [24,25].

###### Micronuclei

Findings from several studies support a model where chromothripsis arises from the disintegration of micronuclei, which are formed from lagging chromosomes caused by mitotic errors [27,28,29]. Micronuclei were shown to be more susceptible to DNA damage than the nucleus [8]. Using a combination of live-cell imaging and single-cell genome sequencing, Zhang et al., demonstrated that micronucleus formation can induce CRs that exhibit all known features of chromothripsis [30].

###### Common Fragile Sites (CFSs)

CFSs are defined as specific genomic regions that are prone to DNA breaks under mild DNA replication stress [31]. To date, more than 100 CFSs have been identified in the human genome [32]. CFSs are hotspots for CRs in cancer [32]. CFSs are stable under normal growth conditions, but become unstable and prone to DNA breaks when the replication process is perturbed [32]. CFSs are characterized by late replication timing and enrichment in large genes [33]. Recent studies have revealed the cell type specificity of CFSs [33]. For example, FRA3B in *FHIT* is unstable in lymphocytes, lymphoblasts, HCT116, and HeLa cells, whereas it is stable in normal fibroblasts [33]. The instability of CFSs may be attributable to their characteristics in DNA sequences, such as AT-rich sequences; however, the exact mechanisms remain unclear [33].

###### Meiosis

Meiosis is a specialized cell division that produces haploid gametes, such as eggs and sperms, from diploid cells [34,35]. It involves one round of DNA replication followed by two rounds of chromosome segregation: meiosis I and meiosis II [34,35]. Homologous chromosomes are separated in meiosis I, then sister chromatids are separated in meiosis II [34,35]. In meiosis I, recombination between maternal and paternal chromosomes occurs, which confers genetic diversity in progeny [34,35]. The meiotic recombination is initiated during early prophase I with DSB induction by Spo11 protein [34,35,36]. Spo11 is an evolutionally conserved enzyme that resembles archaeal topoisomerase VI [34,36]. After DSB induction by Spo11, DSB end resection is carried out by resection factors, such as MRE11 and EXO1, in order to start the recombination reaction [34,36].

###### Non-Physiological Sources of DSBs

Ionizing Radiation (IR).

IR is a potent inducer of CR. IR induces various types of DNA lesions, including base damage, SSBs, and DSBs [37]. Among these, DSBs are the major cause of CR. The frequency of CRs, such as translocations and dicentrics, increases with the dose of IR [38].

###### Anti-Cancer Drugs

Many anti-cancer drugs are meant to kill cancer cells by damaging DNA. Therefore, some of them generate DSBs directly or indirectly. One example is etoposide, which is an inhibitor of Top2 [39]. Etoposide traps Top2cc, the catalytic intermediate of Top2, and inhibits subsequent reaction, resulting in persistence of DSBs [18,39]. Another example is cisplatin, which covalently binds DNA and forms intrastrand- and interstrand DNA crosslinks [40]. Such DNA crosslinks can lead to DSBs, especially when interstrand crosslinks occur in the S phase [41].

#### 2.2.3. DSB Repair Pathways Involved in the Generation of CR

Mammalian cells undertake DSB repair through two major pathways: non-homologous end joining (NHEJ) and homologous recombination (HR) (Figure 4a,c) [42,43]. In NHEJ, the Ku70/80 heterodimer first binds to the DSB ends and holds the two DSB ends in close proximity (Figure 4a) [42,43]. The Ku heterodimer recruits other NHEJ factors, including DNA-PKcs, XRCC4, XLF, PAXX, and LIG4 [42,43]. LIG4 ligates the two broken ends. This type of NHEJ is referred to as “classical NHEJ” (c-NHEJ), in contrast to the other end-joining pathway that is independent of c-NHEJ factors, referred to as alternative NHEJ (a-NHEJ) [42,43]. The factors functioning in a-NHEJ include the MRE11-RAD50-NBS1 (MRN) complex, CtIP, poly(ADP-ribose) polymerase 1 (PARP1), and Polθ [42,43]. The endonuclease activity of the MRN complex, which is promoted by CtIP, initiates a-NHEJ by resecting DSB ends to generate 3′ single-strand DNA (ssDNA) overhangs (Figure 4b) [42,43]. Polθ stabilizes the annealing between homologous sequences on the two 3′ ssDNA overhangs and extends one 3′ DNA end using the annealing partner as a template [42]. Although it is evident that PARP1 promotes a-NHEJ, the underlying mechanism remains elusive. In a-NHEJ, LIG3 or LIG1 seals the broken DNA ends [42]. While HR is also initiated by MRN- and CtIP-dependent resection, this pathway requires extensive resection (>50 nucleotides) compared to that required in a-NHEJ [42,43]. The long-range resection is mediated by exonuclease 1, the endonuclease DNA2, and Bloom syndrome helicase (BLM) (Figure 4c) [42,43]. The resultant ssDNA is rapidly coated with the RPA complex, which is composed of RPA1, RPA2, and RPA3 [43]. The binding of RPA to the ssDNA is considered to limit secondary structure formation and interaction of the ssDNA in other nuclear events, such as nuclease attack [43]. Next, RAD51 recombinase replaces the RPA complex on the ssDNA in a manner dependent on BRCA1, BRCA2, and PALB2 (Figure 4c) [43]. RAD51 helps ssDNA invade the DNA duplex and facilitates base pairing with complementary sequences. Although several subpathways of HR can be initiated after strand invasion, “synthesis-dependent strand annealing” is the predominant pathway existent in somatic cells [43]. 

DNA sequences at translocation breakpoint junctions present in patient-derived tumors suggest that translocations arise via NHEJ, as the sequences at the junctions do not exhibit significant homology [44]. Earlier studies on mouse cells have shown that a-NHEJ was the primary pathway adopted in translocation [45,46,47]. In these studies, gene knockout or RNAi-mediated depletion of a-NHEJ factors, such as CtIP, XRCC1, LIG3, or LIG1, decreased the frequency of translocation after DSB induction by I-SceI endonuclease [45,46,47]. However, a later study performed using genome editing technology demonstrated that in human cells, translocation is primarily formed via XRCC4/LIG4-dependent c-NHEJ, and that a-NHEJ initiates translocation when the c-NHEJ is defective [48]. A recent study using radiation identified a novel NHEJ subpathway named “resection-dependent c-NHEJ” as a pathway involved in translocation during the G1 phase [49]. This pathway requires factors involved in resection, including MRE11, CtIP, EXD2, and Artemis, in addition to c-NHEJ factors, such as DNA-PKcs [49]. Collectively, the findings from these studies suggested that c-NHEJ is the major repair pathway that generates translocation in human cells.

### 2.3. Clinical Significance of CR

From an evolutionary standpoint, CR is beneficial to organisms because it increases genetic diversity. However, CR has also been associated with various diseases. For example, recent studies have established the role of CNVs in the etiology of diseases such as autism spectrum disorders, sporadic schizophrenia, mental retardation, and congenital limb malformation [50,51,52,53,54,55,56,57]. CRs have also been implicated in infertility and miscarriage risk [58]. Moreover, CR is one of the most common events in almost all types of cancer [59]. CR can produce fusion genes, which are identified in many types of hematopoietic malignancies and solid tumors [59,60]. A fusion gene, as the name suggests, comprises two distinct genes fused in frame [59,60]. Fusion genes often consist of protooncogenes, such as *ABL, MYC*, and *ALK* [59,60]. As products of fusion genes, protooncogene products often acquire increased activity or expression levels, thereby promoting leukemogenesis or carcinogenesis [59,60]. For example, in the *BCR-ABL1* fusion gene product, which is commonly found in chronic myeloid leukemia, ABL1 tyrosine kinase is constitutively activated even in the absence of growth-stimulating signals [59]. Another example is the *IGH-MYC* fusion gene in Burkitt’s lymphoma, which drives *c-myc* overexpression owing to the juxtaposition of *MYC* with the strong enhancer of the immunoglobulin (Ig) gene [59]. The examples of fusion genes expressed in hematopoietic malignancies and solid tumors are listed in Table 1.

### 2.4. CRs in A-T Patients

Although the frequency of CR is higher in several types of cells obtained from A-T patients, the most frequent CRs observed are those involving chromosomes 7 and 14 in T lymphocytes. In A-T patients, approximately 10% of all T lymphocytes exhibit inversions and translocations in chromosomes 7 and 14 [4,5]. The breakpoints of these CRs are located in the T cell receptor genes, indicating that the misrepair of DSBs arising during V(D)J recombination causes CR [4,5]. These CRs are also detected in the lymphocytes of healthy individuals, but at a considerably lower frequency [4,5]. Translocations in chromosomes 7 and 14 are estimated to increase by approximately 40-fold in A-T patients compared to that in normal individuals [5]. In contrast to that in T cells, CRs involving Ig genes in B cells have not been reported to increase notably, although B cells also undergo V(D)J recombination [4,5]. Fibroblasts derived from A-T patients exhibit higher rates of CR, but the CRs are not consistent among patients and do not involve chromosomes 7 and 14 or Ig genes [4,5]. A recent whole genome analysis of acute lymphoblastic leukemia demonstrated that chromothripsis was observed at a higher frequency in A-T patients than in patients with other types of DNA repair syndromes [61].

## 3. Mechanism of ATM-Dependent Suppression of CR

### 3.1. ATM and DNA Damage Response (DDR)

Upon DNA damage, especially DSBs, ATM is activated and initiates signaling cascades, which are collectively termed DNA damage response (DDR) [62]. DDR involves cell cycle checkpoint regulation and DNA repair [62]. DNA damage-induced cell-cycle checkpoint control refers to the halting of the cell cycle to prevent progression to subsequent phases when cells undergo DNA damage [63,64]. DNA damage-induced cell-cycle checkpoints include the G1 checkpoint, S checkpoint, and G2 checkpoint; ATM is involved in all the checkpoints (Figure 5a) [63,64].

In G1 checkpoint control, cell cycle progression is halted at the late G1 phase when G1-phase cells undergo DNA damage [63,64]. At this point, ATM directly phosphorylates p53 at the serine-15 residue in its amino-terminal transactivation domain (Figure 5(a-1)) [63,64]. ATM-activated Chk2 kinase also phosphorylates p53 at the threonine-18 and serine-20 residues [63,64]. p53 phosphorylation at these residues activates p53 as a transcription factor [63,64]. In addition, ATM phosphorylates murine double minute 2 (MDM2), which binds and ubiquitylates p53 for proteasomal degradation [63]. The ATM-dependent phosphorylation of p53 and MDM2 inhibits the interaction between these molecules, which stabilizes p53 [63]. The key target of p53 in the G1 checkpoint is the *p21* gene, which encodes the p21/WAF1/CIP1 inhibitor of cyclin-dependent kinase 2 (CDK2) [63,64]. Progression from the G1 to the S phase is promoted by the CDK2-dependent phosphorylation of Rb protein and dissociation of Rb from E2F, which transactivates the genes associated with G1-S progression [63,64]. p21 inhibits CDK2-dependent Rb phosphorylation, thereby maintaining E2F in the inactivated state [63,64].

At the S checkpoint, the firing of uninitiated replication origins is transiently inhibited [63,64]. At least two molecular pathways of S checkpoint control have been identified thus far, both of which are controlled by ATM [63,64]. One pathway operates through the ATM/Chk2/CDC25A signaling cascade. CDC25A is a phosphatase that dephosphorylates CDK2, thereby activating the cyclin E/CDK2 and cyclin A/CDK2 complexes [63,64,65]. CDK2 promotes the loading of CDC45, a factor required for new origin firing, onto the chromatin [63]. In response to DNA damage in the S phase, ATM-activated Chk2 phosphorylates CDC25A (Figure 5(a-2)) [63,64,65]. The phosphorylation of CDC25A promotes ubiquitylation and proteasomal degradation of this protein (Figure 5(a-2)) [63,64]. ATR-activated Chk1 kinase is also known to be involved in CDC25A phosphorylation (Figure 5(a-2,3)) [63,64,65]. The decrease in CDC25A levels suppresses the CDK2 activity, which prevents the initiation of new origin firing [63,64]. The ATM-dependent phosphorylation of NBS1 and SMC1 plays a role in the other S checkpoint control pathway (Figure 5(a-2)) [63,64]. This pathway also depends on BRCA1 and FANCD2 [63,64]. However, it remains unclear how these molecules suppress new origin firing.

At the G2 checkpoint, cells with DNA damage induced in the G2 phase are prevented from entering mitosis. The key target at this checkpoint is cyclin B/CDK1 kinase, which promotes the G2-M transition [63,64]. At the G2-M transition, cyclin B/CDK1 is activated by CDC25 family phosphatases, namely CDC25A, B, and C [63,64,65]. In response to DNA damage in the G2 phase, ATM and ATR phosphorylate Chk2 and Chk1, respectively. In turn, activated Chk2 and Chk1 phosphorylate CDC25 phosphatases, thereby inhibiting or degrading these phosphatases (Figure 5(a-2,3)) [63,64,65].

Following activation in response to the introduction of DSBs, ATM phosphorylates a number of proteins involved in DSB repair. These proteins include MRE11, RAD50, NBS1, BRCA1, and 53BP1 [66]. Indeed, ATM has been shown to be involved in both NHEJ and HR. With respect to NHEJ in the G0/G1 phase, the repair of the majority of radiation-induced DSBs does not require ATM; however, approximately 10–15% of the DSBs are repaired in an ATM-dependent manner [67]. These DSBs are repaired at a slow rate and also require the MRN complex, Artemis, and 53BP1 [68,69]. Previous studies have indicated that these DSBs are present in heterochromatin, which is a compact chromatin structure [70,71]. There is ample evidence indicating that heterochromatin prevents DSB repair [71]. Goodarzi et al., found that ATM plays a role in relaxing heterochromatin, thereby providing the repair factors access to the DSBs in heterochromatin [70,71]. Specifically, in response to DSBs, ATM phosphorylates Krüppel-associated box-associated protein-1 (KAP-1), a heterochromatin-building factor, at serine-824 (Figure 5b) [70,71]. KAP-1 is known to undergo constitutive SUMOylation at several residues in its C-terminal regions, which mediates the interaction between KAP-1 and the SUMO-interacting motif in chromodomain-helicase-DNA binding 3.1 (CHD3.1) [71,72]. CHD3.1 is a subunit of the nucleosome remodeling and deacetylase complex, which is involved in heterochromatin formation [71,72]. Goodarzi et al., demonstrated that the ATM-dependent phosphorylation of KAP-1 at serine-824 interrupted the interaction between SUMOylated KAP-1 and the SUMO-interacting motif of CHD3.1, which resulted in chromatin relaxation [71,72].

In the G2 phase, normal human cells use NHEJ to repair ~70% of the radiation-induced DSBs, whereas the remaining ~30% DSBs are repaired using HR [73]. Beucher et al., demonstrated that ATM, along with Artemis, contributes to the HR of radiation-induced DSBs in the G2 phase [74]. Cells deficient in ATM or Artemis exhibited similar DSB repair defects as those deficient in HR factors, such as BRCA2, in the G2 phase [74]. Moreover, resection and RAD51 loading were impaired in irradiated G2 cells deficient in ATM or Artemis [74]. It is likely that HR repairs DSBs in heterochromatin, because the depletion of KAP-1 relieved ATM dependency for repair [74]. Additionally, ATM directly promotes resection by phosphorylating CtIP (Figure 5b) [75,76]. Therefore, ATM appears to perform dual functions in HR in the G2 phase: heterochromatin relaxation by KAP-1 phosphorylation and promotion of resection by CtIP phosphorylation [76].

### 3.2. Mechanisms Underlying the ATM-Dependent Suppression of CR at Immune Gene Loci

Similar to those in A-T patients, approximately 10% of primary T lymphocytes in ATM^(-/-)^ mice exhibit chromosomal aberrations involving the T cell receptor α loci [77,78,79]. In addition, breaks and translocations associated with immunoglobulin heavy chain (IgH) gene on chromosome 12 accumulated in B cells derived from ATM^(-/-)^ mice [79]. In ATM^(-/-)^ lymphocytes, V(D)J recombination is not abrogated, but the repair of coding ends is significantly impaired, which leads to the formation of aberrant chromosomes, such as terminally deleted chromosomes [78,80,81,82]. These chromosomal aberrations are mainly dependent on RAG1/2 endonucleases, which introduce DSBs in Ig and T cell receptor-encoding gene segments [79]. In B and T cells from ATM^(-/-)^ mice, chromosomes with telomere deletion induced by RAG1/2 can survive several rounds of cell cycle without the induction of DDR [79]. Consistently, the development of thymomas with TCRα translocation in ATM^(-/-)^ mice requires RAG [83]. Bredemeyer et al. demonstrated that ATM promotes c-NHEJ by stabilizing RAG-mediated DSB post-cleavage complexes in V(D)J recombination [80]. Following activation by DSBs, ATM phosphorylates histone H2AX, a subtype of histone H2A, at serine-139 [84]. H2AX phosphorylation initiates a cascade of the assembly of proteins around DSBs [85]. These proteins include MDC1 and 53BP1 [85]. The assembly of these proteins can be visualized as “nuclear foci” using immunofluorescence [85]. Interestingly, H2AX-deficient B cells activated for V(D)J recombination or CSR exhibit higher translocation frequency in Ig genes [86,87,88]. Moreover, when activated for CSR, ATM-, H2AX-, 53BP1-, or MDC1-deficient B cells exhibit translocation involving the IgH gene at a higher frequency than their wild-type counterpart [88]. These findings suggested that ATM-dependent foci factors play significant roles in suppressing translocation in immune cells [87].

### 3.3. Mechanisms Underlying the ATM-Dependent Suppression of Radiation-Induced CR

#### 3.3.1. ATM Suppresses Interchromosomal CR Induced by IR

Unlike the cases in CRs caused by V(D)J recombination and CSR, the mechanisms by which ATM suppresses radiation-induced CR have not been studied extensively. A previous study investigated the mechanism underlying the suppression of radiation-induced CR by ATM [89]. In this study, the authors employed dicentric chromosome as a marker of interchromosomal CR. Similar to translocation, dicentric is a type of interchromosomal SV that occurs when two broken chromosomes are rejoined erroneously (Figure 1) [90]. Translocation and dicentric formation are known to occur at similar frequencies [89,91]. They developed an assay called “translocation assay” to efficiently quantify the frequency of radiation-induced dicentrics. This method includes the synchronization of the cell cycle at the G0/G1 phase, γ-ray irradiation, release from cell cycle synchronization, isolation of mitotic cells, chromosome sample preparation, and detection of dicentrics using centromere/telomere FISH (Figure 6a,b) [89]. With the assay, they found that the number of radiation-induced dicentrics increased in human fibroblasts (BJ-hTERT) when ATM was chemically inhibited or depleted using shRNA [89]. As described above, ATM is involved in both cell cycle checkpoint control and the repair of a subset of DSBs in the G1 phase. To determine the function of ATM responsible for the suppression of CR, they investigated ATM-dependent G1 checkpoint control. Since ATM mediates G1 checkpoint control by activating p53, they examined the effect of p53 depletion on the dicentric frequency. Short-hairpin (sh)RNA-mediated p53 depletion significantly increased the frequency of IR-induced dicentrics, indicating that ATM-dependent G1 checkpoint contributes to the suppression of radiation-induced CR [89]. However, ATM depletion led to a more significant increase in the frequency of dicentrics than that achieved with p53 depletion [89]. Therefore, they next investigated the checkpoint-independent function of ATM in CR suppression. To this end, the translocation assay was performed in the presence of p53 shRNA and Chk1 inhibitor to suppress all cell cycle checkpoints. The experiment revealed that ATM inhibition increased the dicentric frequency even under checkpoint inhibition, indicating that ATM also plays a role in CR suppression independent of its checkpoint regulatory function [89].

#### 3.3.2. ATM Suppresses Pairing between DSBs

CR requires the concurrence of DSBs at two or more genomic regions and misrejoining between them (Figure 7) [10]. The prerequisite for misrejoining between two or more DSBs is that they should be located in close proximity, which is referred to as “DSB pairing” (Figure 7). Although DSB pairing is necessary for rejoining between DSBs at different locations, the underlying mechanism has not been well studied. A study attempted to identify the factors regulating DSB pairing [92]. To visualize DSB pairing in live cells, they used mCherry-BP1-2, which is a fusion protein of mCherry and the minimal focus-forming region of 53BP1 (1220–1711 amino acids) [93]. They generated BJ-hTERT cells stably expressing mCherry-BP1-2 protein and examined how DSB pairing occurs after γ-ray irradiation using live cell imaging. The experiment revealed that the mCherry BP1-2 foci paired dynamically, which indicated that chromatin containing DSBs move and associate dynamically with each other [92]. Notably, the foci pairing frequency was higher in ATM-deficient or inhibited cells, indicating that ATM is involved in the suppression of the pairing between two DSBs [92]. Collectively, their findings indicated that ATM suppresses radiation-induced CR in at least two ways: G1 checkpoint control and suppression of DSB pairing.

## 4. Unresolved Questions and Future Perspectives

Although CR is implicated in various human diseases [51,52,53,54,55,56,57,58], it remains to be determined whether the CR-suppressing function of ATM is associated with the phenotypes observed in A-T patients. It is likely that the CR-suppressing function of ATM is related to the predisposition of A-T patients to cancer and hematopoietic malignancies; however, the relation between the CR-suppressing function and other symptoms is unknown. A-T patients exhibit various clinical manifestations, including neurodegeneration, premature senescence, and infertility [3]. Of note, previous studies have indicated the association between CR and infertility [58]. Recent technical advances in genome structural analysis have revealed that CR alters the three-dimensional structure of the genome, such as topologically associating domains (TADs) [50]. TADs are self-interacting genomic regions recently identified using HiC, a technique that enables the detection of interaction among different genomic regions [94]. Recent studies have suggested that TADs facilitate interaction between regulatory elements (e.g., enhancers and promoters) present within them while restricting regulatory effects from neighboring TADs [50]. CR can disrupt TADs, which cause aberrant gene expression and may potentially induce diseases [50]. Therefore, the comparative analyses of three-dimensional genome structure and transcriptome between normal and A-T cells may contribute to the better understanding of the etiology of the various phenotypes in A-T patients.

## Figures and Tables

**Figure 1 genes-12-01232-f001:**
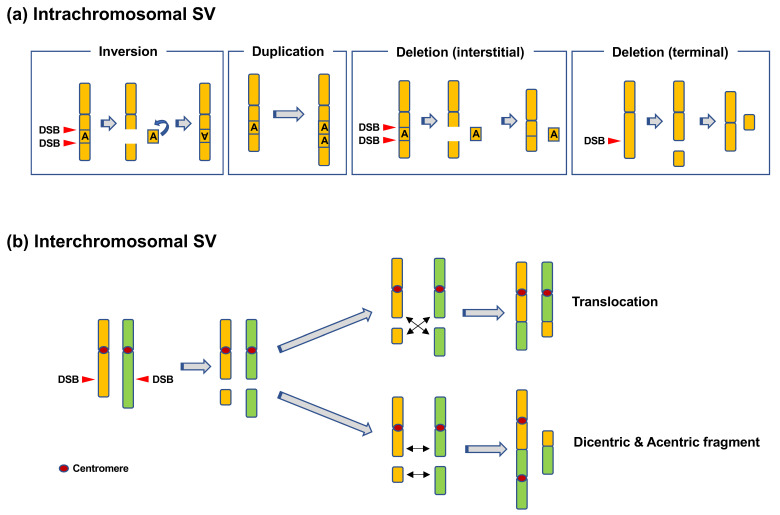
Types of chromosomal rearrangements. (**a**) Intrachromosomal structural variation (SV). Inversions occur when an interstitial fragment formed as a result of two DNA double-strand breaks (DSBs) is inverted and rejoined. Duplications occur when a specific region of a chromosome is tandemly duplicated. Deletions can be categorized as interstitial deletions and terminal deletions. Interstitial deletions occur when two DSBs in a chromosome are rejoined with an interstitial fragment left unrejoined. Terminal deletions occur when a DSB in a chromosome is unrepaired. (**b**) Interchromosomal SV. Translocations occur when two broken chromosomes with and without a centromere are rejoined. Dicentrics are formed when two broken chromosomes with centromeres are rejoined.

**Figure 2 genes-12-01232-f002:**
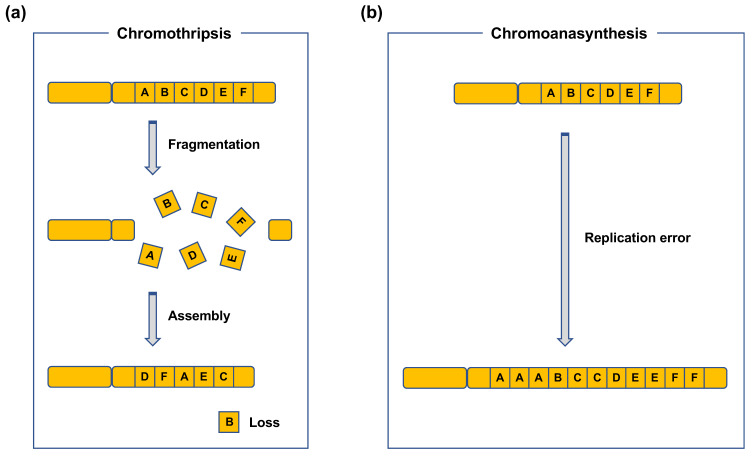
Chromoanagenesis. (**a**) Chromothripsis. Chromothripsis occurs when fragments formed from multiple DSBs introduced in one or a few chromosome(s) are randomly reassembled. (**b**) Chromoanasynthesis. Chromoanasynthesis is characterized by the presence of duplications, triplications, and segment losses frequently localized to one or a few chromosome(s). Chromoanasynthesis is considered to result from a defective DNA replication process, such as template-switching.

**Figure 3 genes-12-01232-f003:**
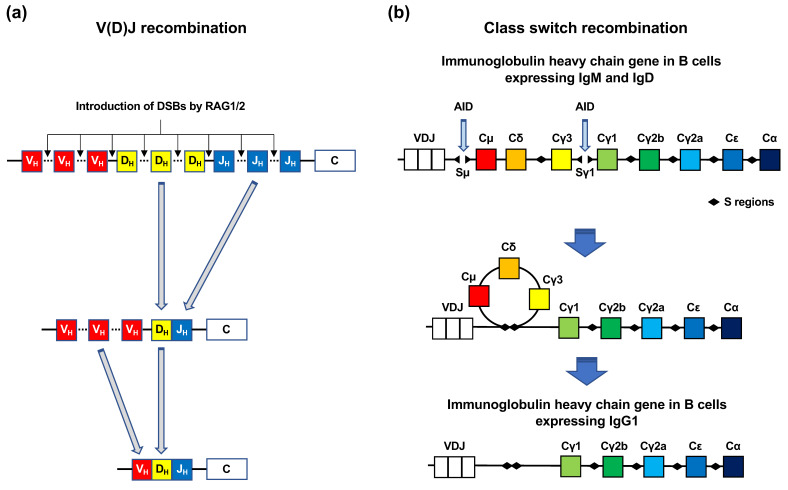
V(D)J recombination and class switch recombination. (**a**) V(D)J recombination. Immunoglobulin (Ig) genes in B cells and T-cell receptor genes in T cells have variable (V), diversity (D), and joining (J) gene segments. Each of the V, D, and J segments contain multiple exons. The RAG1/2 complex introduces DSBs at the recombination signal sequences localized between the exons of V, D, and J segments. First, an exon of the D segment and an exon of the J segment are selected and rejoined, following which an exon of the V segment is selected and rejoined to the D-J segment. (**b**) Class switch recombination (CSR). CSR to IgG1 is shown. Before CSR, the Ig heavy chain (IgH) gene expresses IgM and IgD. Upon CSR activation, activation-induced cytidine deaminase (AID) deaminates the Sμ and Sγ1 regions, which results in DSB formation in the S regions. The Sμ and Sγ1 regions are rejoined by intrachromosomal deletion, which facilitates the expression of the IgG1 gene.

**Figure 4 genes-12-01232-f004:**
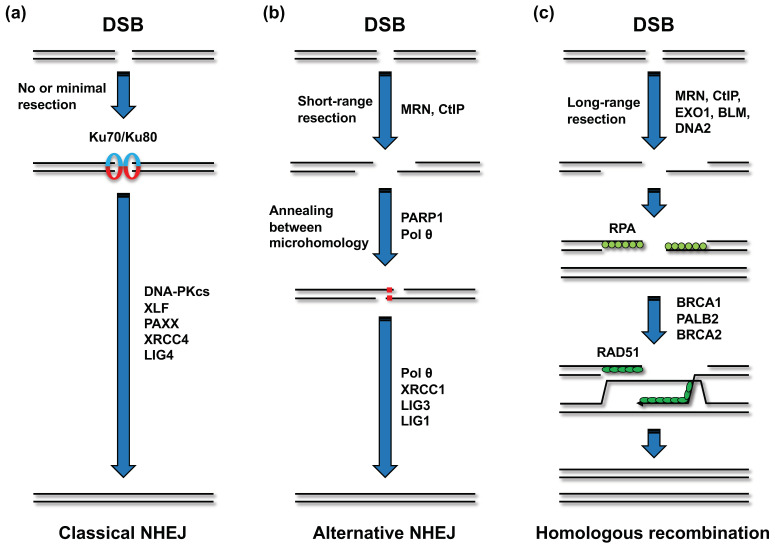
DSB repair pathways. (**a**) Classical non-homologous end joining (NHEJ). In classical NHEJ, the Ku70-Ku80 heterodimer binds to the DSB ends. The DSB-bound Ku heterodimer recruits other classical NHEJ factors, including DNA-PKcs, XLF, PAXX, XRCC4, and LIG4, and at the final step, LIG4 ligates the two DSB ends. (**b**) Alternative NHEJ. Alternative NHEJ is initiated by short-range resection (<25 bp). The MRE11-RAD50-NBS1 (MRN) complex and CtIP are involved in the resection process. PARP1 and DNA polymerase θ (Polθ) promote the subsequent reaction. In the case of alternative NHEJ, LIG3 or LIG1 seals the annealed intermediate with assistance from XRCC1. (**c**) Homologous recombination (HR). HR requires a longer range of resection than alternative NHEJ. Long-range resection is conducted by EXO1, BLM, and DNA2 along with the MRN complex and CtIP. The resection generates a single-strand DNA that is immediately coated with RPA. Following this, RPA is replaced by RAD51 in a manner dependent on BRCA1, PALB2, and BRCA2. RAD51 promotes the invasion of the single-strand DNA into the sister chromatid, followed by homology search. Although several subpathways can operate after strand invasion, “synthesis-dependent strand annealing” is the predominant pathway in somatic cells.

**Figure 5 genes-12-01232-f005:**
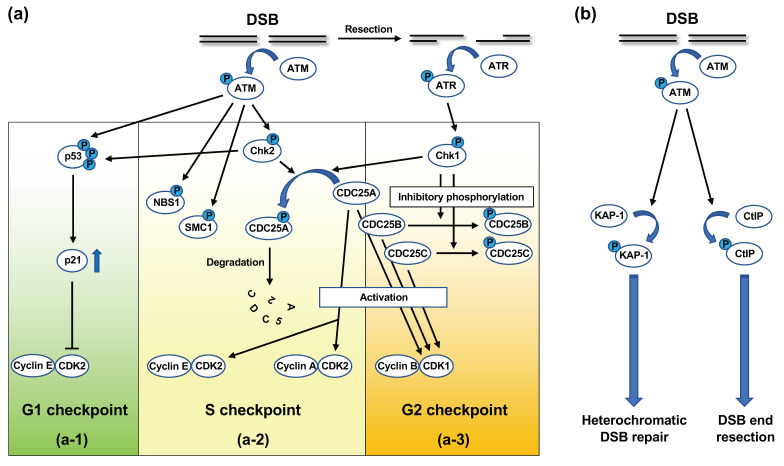
ATM-dependent signaling in response to DSBs. (**a**) ATM-dependent signaling in cell cycle checkpoint. (**a-1**) G1 checkpoint. In unperturbed cells, cyclin E/CDK2 promotes G1-S transition. In response to DSBs arising in the G1 phase, activated ATM and ATM-activated Chk2 phosphorylate p53, a transcription factor, which enhances the stability and activity of p53. Activated p53 induces the expression of p21, which inhibits the activity of CDK2, thereby blocking cell cycle progression at the late G1 phase. (**a-2**) S checkpoint. Cyclin E/CDK2 and cyclin A/CDK2 promote G1-S transition and origin firing, respectively. CDC25A is a phosphatase that activates CDK2 via dephosphorylation. In response to DSBs arising in the S phase, ATM activates Chk2 by phosphorylation. In turn, activated Chk2 phosphorylates CDC25A, which promotes the ubiquitylation and proteasomal degradation of CDC25A. The resection of the DSB ends activates ATR, which then activates Chk1 via phosphorylation. Chk1 can also phosphorylate CDC25A. The ATM-dependent phosphorylation of NBS1 and SMC1 is suggested to play a role in the S checkpoint; however, the exact mechanism remains unknown. (**a-3**) G2 checkpoint. Cyclin A/CDK2 and cyclin B/CDK1 promote G2-M transition. CDC25 phosphatases (CDC25A, B, C) activate CDK1 via dephosphorylation. In response to DSBs arising in the G2 phase, ATM-activated Chk2 and ATR-activated Chk1 phosphorylate CDC25 phosphatases, thereby suppressing their function. (**b**) ATM-dependent signaling in DSB repair. In response to DSBs, ATM phosphorylates KAP-1, a heterochromatin-building factor. The KAP-1 phosphorylation leads to chromatin relaxation, thereby facilitating DSB repair in heterochromatin. ATM also phosphorylates CtIP, a critical resection factor, and promotes DSB end resection, which is critical for resection-dependent DSB repair pathways, such as homologous recombination.

**Figure 6 genes-12-01232-f006:**
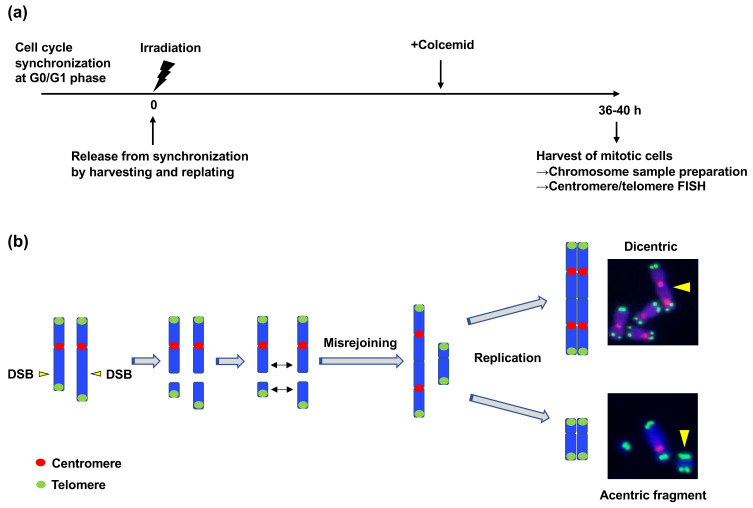
The translocation assay. (**a**) Schematic representation of the translocation assay. First, the cell cycle was synchronized at the G0/G1 phase via contact inhibition. Next, cells were irradiated with γ-rays and released from synchronization by replating at low density. Colcemid was added at the appropriate time point after replating (depending on the experiment) to increase mitotic cells. Thirty-six to forty hours after replating, the mitotic cells were harvested, treated with hypotonic buffer, and fixed. The fixed cells were dropped onto slide glasses, dried, and subjected to fluorescence in situ hybridization using centromere and telomere probes. (**b**) Dicentric formation. DSBs on two different chromosomes are required for the formation of a dicentric. A dicentric is formed when two broken chromosomes with centromeres are rejoined. Dicentric is categorized as a chromosome-type aberration that is formed before DNA replication. The formation of acentric fragments accompanies dicentric formation.

**Figure 7 genes-12-01232-f007:**
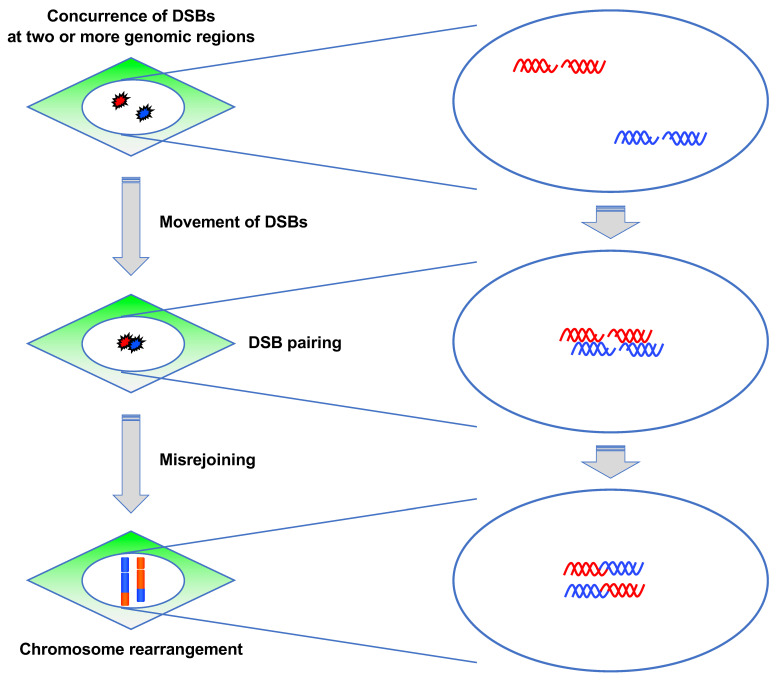
Steps of chromosome rearrangement. For chromosome rearrangements to occur, DSBs need to be introduced concurrently at two or more genomic locations. Following this, the DSBs position themselves in close proximity and are misrejoined.

**Table 1 genes-12-01232-t001:** Fusion genes in hematopoietic malignancies and solid tumors. Typical fusion genes associated with each disease are listed.

Fusion gene	Disease
***BCR-ABL1***	**Chronic myeloid leukemia**
***IGH-MYC***	**Burkitt lymphoma**
***RUNX1–RUNX1T1***	** Acute myeloid leukemia**
***IGH–MAF***	**Multiple myeloma**
***PML–RARA***	**Acute promyelocytic leukemia**
***EWSR1-FLI1***	**Ewing sarcoma**
***JAZF1–PHF1***	**Endometrial stromal sarcoma**
***PAX3–FOXO1A***	**Rhabdomyosarcoma**
***ETV6-NTRK3***	**Breast carcinoma**
***ALPHA–TFEB***	**Kidney carcinoma**
***TMPRSS2-ERG***	**Prostate carcinoma**
***RET-CCDC6***	**Thyroid carcinoma**
***BRD4–NUT***	**Aggressive midline carcinoma**
***MECT1–MAML2***	**Mucoepidermoid carcinoma**

## Data Availability

Not applicable.

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
