# Peer review of "Mechanisms Underlying the Suppression of Chromosome Rearrangements by Ataxia-Telangiectasia Mutated"

_genes, 2021, doi:10.3390/genes12081232_

Round 1

Reviewer 1 Report

The review by Yamamuchi M. entitled “Mechanisms Underlying the Suppression of Chromosome Rearrangements by Ataxia-Telangiectasia Mutated” aims to give an overview on the current knowledge related to the mechanicistic role of ATM on suppression of chromosome rearrangements.

Overall, this is a comprehensive and useful work, relevant for cancer biology highlighting the key role of ATM in genomic stability and in Ataxia-telangiectasia disease.

The review is well written and clear, the specific critical role of ATM in cell cycle checkpoint regulation and DNA damage repair through homologous and non-homologous recombination after physiological and non-physiological DNA damage are extensively described. Therefore, I recommend this review for publication in "Genes" journal after minor revision.

As a general comment the review is divided in two parts, one more general, related to chromosome rearrangement sources and mechanisms and the other more specific, related to ATM functions in the DNA damage response, that should be more integrated each other in line with the title.

Physiological DNA damage should be specified and differentiated from non-physiological DNA damage.

Specific comments:

  • Line 25: ATM activation by intermolecular autophosphorylation is still debated  (see Lee J-H and Paull T, 2005; Pellegrini M et al., 2006; Daniel J et al., 2008).
  • Line 43: “CR” should be written in full.
  • Line 52: check “copy-neutral…..deletions”.
  • Line 101: the caption 2.3.1 can be removed.
  • Line 104: DNA damage initiated by Spo11, occurring during meiosis, should be also mentioned in the “Sources of DSBs”.
  • DSBs generated by genotoxic agents should be included and commented as non-physiological sources of DSBs.
  • Figure 5 should be split in two parts for DNA damage repair and cell cycle checkpoint.
  • Check line 329 for typos.
  • The caption 4.2 should mention to:

1 The IgH-associated breaks and translocations on Chr12 accumulated at high levels in Atm−/− B cells (for references Callen E et al .,Cell 2007 “ATM Prevents the Persistence and Propagation of Chromosome Breaks in Lymphocytes”).

2 The potential structural role of Atm, during NHEJ, and the kinase activity role, during HR, as suggested by the kinase dead models (Daniel J et al., 2012; Yamamoto K et al., 2012).

  • Figure 7 should be improved.

Author Response

Dear Reviewer 1

Thank you so much for giving me good suggestions. I addressed your comments point by point, except for specific comment #9-2. I would be grateful if you could give me some advice about this point.

General comment #1

As a general comment the review is divided in two parts, one more general, related to chromosome rearrangement sources and mechanisms and the other more specific, related to ATM functions in the DNA damage response, that should be more integrated each other in line with the title.

→I changed the structure of the manuscript so that the two parts are more integrated. The new structure of the manuscript is as follows (the changes are indicated as bold characters):

Abstract

  1. Introduction
  2. Chromosome rearrangement (CR)
    • Types of CR
    • Mechanisms of CR
      • Sources of DSBs
      • DSB repair pathways involved in generation of CR
    • Clinical significance of CR
    • CR in AT patients
  3. Mechanism of ATM-dependent suppression of CR
    • ATM and DNA damage response
    • Mechanisms of ATM-dependent suppression of CR at immune gene loci
    • Mechanisms of ATM-dependent suppression of radiation-induced CR
      • ATM Suppresses Interchromosomal CR Induced by IR
      • ATM Suppresses Pairing between DSBs
  1. Unresolved questions and future perspective

General comment #2

Physiological DNA damage should be specified and differentiated from non-physiological DNA damage.

→I specified physiological sources of DSBs and differentiated them from non-physiological sources of DSBs.

Specific comment #1

Line 25: ATM activation by intermolecular autophosphorylation is still debated  (see Lee J-H and Paull T, 2005; Pellegrini M et al., 2006; Daniel J et al., 2008).

→I deleted “by intermolecular autophosphorylation at serine-1981 and monomerization [3]. Upon activation,” at line 25-26, and connected to the next sentence. The revised sentence is as follows:

In response to DSBs, ATM kinase is activated and phosphorylates proteins involved in cell cycle checkpoint regulation and DNA repair, thereby initiating the response to DNA damage [2].

Specific comment #2

Line 43: “CR” should be written in full.

→ I changed “CR” at line 43 to “Chromosome rearrangement (CR)”.

Specific comment #3

Line 52: check “copy-neutral…..deletions”.

→ It was a simple mistake. I corrected it as follows:

copy-neutral SVs, such as inversions and translocations.

Specific comment #4

Line 101: the caption 2.3.1 can be removed.

→ I removed the caption 2.3.1.

Specific comment #5

Line 104: DNA damage initiated by Spo11, occurring during meiosis, should be also mentioned in the “Sources of DSBs”.

→ I added the description about Spo 11-induced DSBs in meiosis in the “Sources of DSBs”.

Specific comment #6

DSBs generated by genotoxic agents should be included and commented as non-physiological sources of DSBs.

→ I added the section of “non-physiological sources of DSBs” in the “Sources of DSBs”, and included ionizing radiation and anti-cancer drugs.

Specific comment #7

Figure 5 should be split in two parts for DNA damage repair and cell cycle checkpoint.

→ I split Figure 5 into two parts: Figure 5A Cell cycle checkpoint and Figure 5B DSB repair. I also added some sentences to the legend for Figure 5B .

Specific comment #8

Check line 329 for typos (in the legend of Figure 5)

→ In the original manuscript submitted, there are no such typos.

   I corrected them to: In unperturbed cells, cyclin E/CDK2 promotes G1-S transition.

Specific comment #9

The caption 4.2 should mention to:

1 The IgH-associated breaks and translocations on Chr12 accumulated at high levels in Atm−/− B cells (for references Callen E et al .,Cell 2007 “ATM Prevents the Persistence and Propagation of Chromosome Breaks in Lymphocytes”).

2 The potential structural role of Atm, during NHEJ, and the kinase activity role, during HR, as suggested by the kinase dead models (Daniel J et al., 2012; Yamamoto K et al., 2012).

→As for the comment 1, I added the description you suggested to the caption 3.2. (caption 4.2. is changed to caption 3.2 in the revised manuscript). Regarding the comment 2, I appreciate the importance of their finding that kinase-dead mutation of ATM causes embryonic lethality and shows more severe HR defect than null mutation. However, I ‘m afraid that it is quite difficult to discuss this issue in this caption because of the following reasons:

  • The caption 4.2 (caption 3.2 in the revised manuscript) deals with “Mechanisms Underlying the ATM-Dependent Suppression of CR at Immune Gene Loci”.
  • The papers (Daniel J et al., 2012; Yamamoto K et al., 2012) do not show data about CR at immune gene loci.

I would be grateful if you could give me your opinion about whether I should discuss ATM’s role in CR based on the two articles.

Specific comment #10

Figure 7 should be improved.

→ I changed Figure 7 so that the movement and the pairing of two DSBs are more easily understood.

Please find attached the file for the detailed revision.

Sincerely,

Motohiro Yamauchi, Ph.D

Hospital Campus Laboratory, Radioisotope Center, 

Central Institute of Radioisotope Science and Safety,

Kyushu University

3-1-1 Maidashi, Higashi-ku, Fukuoka 812-8582, Japan

Tel: +81-92-642-6194

E-mail: yamauchi.motohiro.619@m.kyushu-u.ac.jp

Best regards,

Motohiro

Reviewer 2 Report

I believe author has a good understanding of the subject and has succinctly  expressed his thoughts in this review. 

Author Response

Thank you for your comments.